# Gender-Specific Patterns of Muscle Imbalance in Elite Badminton Players: A Comprehensive Exploration

**DOI:** 10.3390/sports11090164

**Published:** 2023-08-30

**Authors:** Qais Gasibat, Borhannudin Abdullah, Shamsulariffin Samsudin, Dan Iulian Alexe, Cristina Ioana Alexe, Dragoș Ioan Tohănean

**Affiliations:** 1Department of Sports Studies, University Putra Malaysia UPM, Serdang 43400, Malaysia; drqaiss9@gmail.com (Q.G.); shamariffin@upm.edu.my (S.S.); 2Department of Physical and Occupational Therapy, “Vasile Alecsandri” University of Bacău, 600115 Bacău, Romania; 3Department of Physical Education and Sport Performance, “Vasile Alecsandri” University of Bacău, 600115 Bacău, Romania; alexe.cristina@ub.ro; 4Department of Motric Performance, “Transilvania” University of Brașov, 600115 Brașov, Romania; dragos.tohanean@unitbv.ro

**Keywords:** shoulder muscle imbalance, elites, badminton, asymmetry, shoulder injuries, external and internal ratios

## Abstract

The high-intensity demands of overhead sports exert significant stress on the bilateral shoulder complex, triggering adaptive kinematics and a distinct strength imbalance between internal and external rotators. The imbalance being referred to in the given statement poses a potential risk for humeral head displacement and puts nearby tendons under tension, heightening the vulnerability to injury. This study aims to assess muscle imbalances in badminton athletes. The first hypothesis (H1) suggests that there are differences in internal and external shoulder rotation movements between dominant and non-dominant segments in badminton players. The second hypothesis (H2) proposes that there are variations in muscle imbalances based on gender among elite badminton players. The objectives are to analyze these differences and explore potential gender-related variations in muscle imbalances. The study seeks to contribute to the understanding of muscle imbalances in badminton athletes and potentially guide training and injury prevention strategies in the sport. Using a cutting-edge Hand-Held Dynamometer (HHD), a cohort of 30 elite badminton players underwent an assessment to uncover any bilateral shoulder rotation strength imbalances during a challenging five second isometric maximum contraction. The participants boasted an average age of 17.4 years and a mean playing experience of 7.23 years. The study revealed a notable difference in the ratio of external and internal strength between the dominant and non-dominant shoulders (*p* = 0.000). This discrepancy amounted to a striking 27.93% muscle imbalance in external rotation/internal rotation strength ratios, favoring the dominant shoulder. Moreover, gender-specific differences were detected, with male players exhibiting a 24.54% muscle imbalance in favor of the dominant shoulder, while female players showcased a more substantial 31.33% imbalance (*p* = 0.000). In light of these findings, it became evident that elite badminton players possess considerably stronger dominant shoulders compared with their non-dominant counterparts. Furthermore, the study revealed that male players experience less muscular imbalance than their female counterparts.

## 1. Introduction

Athletes who perform overhead actions during their practices and competitions often experience acute or chronic discomfort [1]. Muscle imbalances can result from discomfort, particularly among individuals engaged in asymmetrical sports such as water polo, squash, and badminton [2]. Badminton, in particular, requires repetitive short-term maximal or submaximal efforts with brief rest intervals [1,3]. Unfortunately, muscle imbalances have been linked to injuries due to disparities in strength and suboptimal performance by athletes [4].

It has been observed that the dominant shoulder of an overhead athlete differs from their non-dominant shoulder in terms of strength, morphology, muscular balance, and range of motion [5]. Studies have shown that most water polo players have stronger dominant shoulders, which creates an imbalance and increases the risk of shoulder injuries for these athletes [1]. Meanwhile, handball players with muscular strength imbalance have a 2.57 times higher relative risk of experiencing shoulder injuries (95% CI: 1.60–3.54; *p* < 0.05) [6]. Higher levels of side-to-side asymmetry are also associated with an increased risk of injuries, as well as the occurrence of postural issues and Low Back Pain (LBP) [7]. In the case of tennis players, their non-dominant side typically has more weakness in lateral flexors and rotator strength compared with physically inactive individuals [8]. However, when comparing tennis players with and without current LBP, there is minimal difference in the balance of trunk strength.

Athletes are at risk of injury due to muscle imbalances between their dominant and non-dominant arms. Sung et al. [9] found that the dominant arm exerts almost 10% more muscle force than the non-dominant arm. In badminton players, the rotator cuff muscles and tendons in the dominant arm are stronger than those in the non-dominant arm [10]. Badminton players may experience shoulder symptoms due to an imbalance in eccentric antagonist and concentric agonist strength [11,12]. Another contributing factor to shoulder injuries is an imbalance in muscular strength between the internal and external rotator muscles [13]. Codine et al. [14] conducted a study to investigate the relationship between agonist and antagonist muscles and their implications for understanding the pathological condition of the joint.

In the context of sport activities such as badminton, several researchers have dedicated their efforts to exploring the variations in shoulder strength and mobility between the dominant and non-dominant sides [15,16,17,18,19,20,21,22]. One study’s findings suggest that the dominant arm tends to have weaker external rotators, stronger internal rotators, and reduced mobility when compared with the non-dominant arm [23]. Moreover, the dominant arm shows lower ratios of concentric external/internal and eccentric external/concentric internal strength. These imbalances in strength ratios during activities such as spiking or throwing may result in acute anterior displacement of the humeral head and increased strain on the biceps’ long-head tendon [23].

During prolonged badminton training, an imbalance between agonist and antagonist muscles can occur, characterized by a greater increase in strength of the shoulder’s internal rotators compared with its antagonistic muscles [24]. Athletes should be acknowledged that there could be notable discrepancies in the strength of the internal and external rotation of their dominant and non-dominant shoulders, which can have implications for shoulder stability and injury prevention [25]. Hadzic et al. [26] reported that an overhead athlete is at an increased risk of injury if their dominant shoulder’s internal rotation is 9 percent stronger than their non-dominant shoulder and their dominant shoulder’s external rotation is 14 percent stronger than their non-dominant shoulder. The ideal strength values may vary depending on factors such as an individual’s specific sport, training regimen, and individual characteristics.

According to Thompson et al. [27], assessing the strength of the shoulder’s internal rotation/external rotation is useful, as these muscle groups are responsible for the dynamic stabilization of the glenohumeral joint. Studies have demonstrated that imbalances between internal rotation and external rotation strength ratios can lead to sports injuries, particularly in activities such as handball [28] and volleyball [29] that involve overhead throwing exercises. To reduce the chances of shoulder injuries, badminton players should focus on avoiding muscle imbalance and properly managing their training. It is important to understand the level of imbalance and effectively control the training load. Evaluating the strength of the shoulder rotators can be helpful in preventing injuries.

To evaluate differences in muscle strength between an individual’s dominant and non-dominant sides, it is customary to assess side-to-side strength asymmetry, as previously documented in studies involving a cohort of twenty healthy adult males with a mean age of 24.65 ± 2.4 years [30,31].

To substantiate the differences and understand their significance, in our study, we implemented an independent *t*-test and compared the movements of the dominant and non-dominant shoulders of elite badminton athletes. The severity and/or long-term duration of a shoulder injury (muscle atrophy and/or disuse) may be responsible for a weakness in strength [6].

This study aims to determine muscle imbalances in badminton athletes. Hypothesis number 1 (H1) assumes the existence of differences between internal rotation and external shoulder rotation movements, between dominant and non-dominant segments, in badminton athletes. To obtain additional information, we propose a second hypothesis (H2), according to which we assume that there are differences in muscle imbalances between elite players according to gender (male and female).

## 2. Materials and Methods

### 2.1. Study Design

In this study, badminton players’ shoulder muscles underwent a unique evaluation using the cutting-edge Microfet 2^TM^ dynamometer. The focus was on both dominant and non-dominant shoulders, uncovering their isometric strength. To determine the isometric strength of the internal and external shoulder muscles, researchers measured the maximum achievable force during a gripping 5 s isometric contraction.

### 2.2. Participants

Conducting a cross-sectional study, this research aimed to explore the physical capabilities of a select group of young badminton elites from the Voltrex Badminton Academy in Kuala Lumpur, Malaysia. The estimated sample size was based on the literature, suggesting that a minimum of 19 participants would be required to determine an intraclass correlation coefficient (ICC) score of 0.7 with two ratters [15]. The study included a total of 30 players, comprising 15 males and 15 females, who showcased remarkable talent and competitive prowess. With equal representation from both genders, the participants were aged around 17 years old (see Table 1) A total of 20 players were right-handed, with only 10 players being left-handed. In terms of playing categories, 15 players participated in singles matches, 8 players in doubles matches, and only 7 players in mixed doubles. These participants were selected based on specific criteria: (1) being competitive elite players; (2) engaging in frequent training and maintaining a training frequency of no more than four times a week; (3) meeting the requirements of the Physical Activity Readiness Questionnaire (2022 PAR-Q+); (4) being free from injury. These highly skilled individuals, with a wealth of experience in the sport, had an average of 7.23 years of training and expertise, demonstrating their extensive knowledge and proficiency in the game of badminton.

Each participant received a comprehensive explanation before voluntarily providing written consent to partake in this ground-breaking research. Upholding ethical standards, the study was conducted with the formal approval of the host institution’s Ethics Committee (JKEUPM-2022-007), strictly adhering to the principles outlined in the 1975 Helsinki Declaration. 

### 2.3. Procedures

In order to evaluate the secrets of shoulder strength, the Microfet 2^TM^ Hand-Held Dynamometry (HHD) (Hoggan Health, Draper, UT, USA) was brought into play, with a measurement range of 0–300 NEWTONS and accuracy within 1% of reading. With the expertise of an experienced assessor proficient in its techniques, this powerful tool effectively gauged the internal and external rotation strength of the participants’ shoulders, supported by a secondary assessor with over three years of experience operating the device.

Before diving into the assessments, the participants were required to complete a questionnaire, providing insights into factors such as hand dominance and any prior shoulder discomfort or issues. Equipped with the Microfet 2^TM^, the participants embarked on a series of bilateral internal and external rotation strength tests, expertly guided by the tester. Before the tests, a five-minute warm-up session focused on shoulder mobility drills, mirroring the directions of the forthcoming measurements.

During the tests, the participants assumed a specific posture with their elbows extended and bent at a 90-degree angle, creating a foundation for accurate assessments (see Figure 1 and Figure 2). The tester played a crucial role, firmly supporting the humerus against the stretcher, while the participants used their opposite arm to lend additional support. By carefully locating anatomical landmarks on the forearm’s posterior surface, the HHD (placed just posterior to the ulnar styloid process) accurately measured the strength of external rotation. For internal rotation strength, the HHD was positioned on the forearm’s anterior surface, proximal to the ulnar styloid process [32].

The tests themselves involved a momentous challenge: each participant had to exert a maximal isometric contraction for five seconds. Following each strength test, there was a well-deserved 60 s break, ensuring the participants could recover. To maintain accuracy and preserve energy, a brief 10 s rest period was granted after every two repetitions. Throughout the arduous process of creating force, the tester provided verbal encouragement, inspiring the participants to push their limits [32].

To analyze the results, the highest scores from each test session were meticulously considered, highlighting the participants’ peak performance. This comprehensive approach aimed to reveal the true strength residing within their shoulders. To assess muscle imbalance, external/internal ratios (ER/IR × 100%)were calculated, including Dominant Ratio (DR) and the Non-Dominant Ratio (NDR). This study promises to assess the intriguing dynamics of shoulder strength in badminton players. To address the first hypothesis (H1), we engaged in the evaluation and analysis of specific movements and, thus, according to the data presented in the tables, we highlighted the existence of differences between the dominant and non-dominant internal and external rotation of the shoulder among our participants.

### 2.4. Statistical Analyses

The independent samples *t*-test is a statistical test used to compare the means of two independent groups or samples. It is often used in scientific research to determine whether there is a significant difference between the means of two groups. The independent samples *t*-test assumes that the two samples are independent of each other and that they are normally distributed. The null hypothesis of the independent samples *t*-test is that there is no significant difference between the means of the two groups, while the alternative hypothesis is that there is a significant difference [33]. In summary, the independent samples *t*-test is a statistical test used to compare the means of two independent groups or samples [34]. It involves stating null and alternative hypotheses, collecting data, calculating the pooled standard deviation and the t-statistic, and comparing it to a critical value to determine whether the null hypothesis should be rejected or not [35].

## 3. Results

To address the first hypothesis (H1), we engaged in the evaluation and analysis of specific movements and, thus, according to the data presented in the tables, we highlighted the existence of differences between the dominant and non-dominant internal and external rotation of the shoulder among our participants.

The data showed us a muscle imbalance of 27.93% in favor of the dominant shoulder. This strong contrast in internal and external rotation highlighted the significant difference in strength between the dominant and non-dominant shoulders in the subjects included in the study. The data presented in Table 2 and Table 3, along with Figure 3, provide numerical and visual evidence of the significant differences observed in muscle imbalance. Specifically, the motor activity of the dominant shoulders of the players included in the study was found to be higher compared with the activity generated by the non-dominant shoulders. These findings strongly support and confirm hypothesis number 1 (H1) regarding the existence of differences in internal and external shoulder rotation movements between dominant and non-dominant segments.

To deepen another aspect of the differences between badminton players, as part of our research approach, we incorporated a secondary investigation to examine muscular imbalances among young male and female elite badminton players. 

The analysis and processing of the data showed us significant differences (*p*-value of 0.00, below the threshold established according to the statistical analysis at *p =* 0.05) between male badminton players and female badminton players regarding muscle imbalances.

In male badminton players, a muscle imbalance of 24.54% in favor of the dominant shoulder was found, while in female badminton players, statistical calculations indicated values of 31.33%, highlighting a much more obvious muscle imbalance.

Based on the analysis of average values, it can be observed that the male badminton players in the study exhibited muscle imbalances at the shoulder level. However, these imbalances were 6.79% lower compared with those observed in the female badminton players, thus confirming hypothesis number 2 (H2). The supporting data for these findings can be found in Table 4 and Table 5, as well as in Figure 4.

The study’s intriguing outcomes shed light on the intricate dynamics of shoulder movement in elite badminton players. The disparities uncovered between dominant and non-dominant shoulders and the contrasting muscle imbalances between genders offer valuable insights into the physical capabilities of these athletes. This research promises to reshape our understanding of shoulder strength in the context of badminton and opens up new avenues for training and performance enhancement.

## 4. Discussion

With the overall objective of determining differences in shoulder strength performance among badminton players, in this present study, the researchers sought to assess asymmetries in shoulder internal and external rotator strength and to compare strength levels and asymmetries between male and female players. The examination and analysis of the data unveiled notable variations in upper limb strength among the badminton players included in the study. The analysis of the expression aspects of the activity of the interested segments related to gender (male–female) showed us that the strength of the internal rotator of the dominant shoulder showed a notable difference, while the asymmetries in the dominant/non-dominant and external/internal ratios did not show a variation by sex. The results indicate that there may be complicated and different dynamics of shoulder force manifestation in young badminton player subjects, suggesting unique patterns and potential influences that shape their physical capabilities.

In researching the field specific to badminton, we considered such an approach new and lacking sufficient and clear data to date. While Petrinović et al. [36] delved into the field of bilateral anthropometric asymmetries in young badminton players, the assessment of shoulder internal and external rotation force production capacities remained an insufficiently analyzed territory, especially in the light of the comparative analysis of male and female players. Starting from this aspect, this study aimed to bring new and substantiated information regarding the different manifestations of shoulder strength in older badminton badminton players. The results revealed significant differences between the dominant and non-dominant shoulder, with a significant difference of 25.4% in both internal and external rotator strength. Previous studies, which analyzed badminton players, reported asymmetries ranging from 9% [37], 10% [38], 14.5% [39], and up to 21.4% [12] in (older) badminton players. Interestingly, these values were almost half of what was observed in tennis players of a similar age [8], highlighting the unique characteristics of badminton athletes. In this perspective, exploring these strength asymmetries helps deepen the understanding of the distinct manifestation of badminton players’ physical capabilities, facilitating that this study corroborated previous findings that players involved in racquet sports possess greater strength in both the internal rotator and external rotator muscles, especially on their dominant side compared with their non-dominant side understanding of factors that might shape performance in this sport [12].

Analyzing the symmetry of shoulder force manifestation in the two genders (male and female), the data indicated the following aspect: a significant difference was observed in the “internal rotators” variable. Although this study represents another attempt to compare the direct display of power between the sexes, we could consider that these differences could be caused by the distinct playing styles adopted by men and women. The specialized literature allows us access to other studies in the field targeted by our research; this aspect indicated the fact that Lee et al. [40] found significant differences in the percentage of shots used in different areas of the court and the types of shots used by men and women. In addition, Blomqvist et al. [41] highlighted that boys between the ages of 12 and 13 had the ability to hit with greater force than girls due to their superior or technical skills and different levels of force expression. This power deficit among women is directly reflected in their game as a higher error rate in the execution of the Smash technique (referring to a powerful overhead shot where the player forcefully hits the shuttlecock downward towards the opponent’s court) that could be attributed to insufficient strength or power. These insights into the relationship between gender, playing style, and strength levels provide insight into the complex interplay of factors that shape the performance of badminton players.

Muscle asymmetries can pose a significant risk to both novice and elite badminton players, increasing the likelihood of injuries. This aspect, supported by research conducted by Dabholkar et al. [42], highlights the urgent need for preventive measures within this population. Movement patterns and the demands placed on the body during badminton play further contribute to the occurrence and development of injuries, as noted by Phomsoupha et al. [43]. To mitigate these risks, it becomes imperative to develop targeted strategies that focus on strengthening and improving techniques [44].

Implementing preventive programs designed specifically for young badminton players holds tremendous potential in reducing the occurrence of injuries. Such initiatives would address the root causes behind different types of injuries. Back injuries, often attributed to a lack of strength, could be minimized through targeted strength-building exercises. On the other hand, arm and shoulder injuries, commonly caused by faulty technique, would benefit from interventions that improve technique and ensure proper movement patterns [44].

While the importance of injury prevention is clear, further research is warranted to evaluate the effectiveness of programs to evaluate and reduce bilateral asymmetries in strength and technique among young badminton players. By continuously investigating, we can gain a better understanding of how these preventative measures affect the health and overall well-being of these talented athletes. Ultimately, this helps to ensure their safety and success.

### 4.1. Limitations of the Study

When conducting a study, it is crucial to recognize the limitations that come with the results. This specific study has a few limitations that should be noted. The sample size was small, as only 30 young elite badminton players from a single academy in Malaysia were included. While their perspectives are valuable, it is important to be careful when applying these findings to other populations with different backgrounds and characteristics.

Another important point is that the study only examined the isometric strength of the internal and external shoulder muscles. This limited focus may not capture all of the factors that contribute to muscle imbalances and the risk of injury. It did not consider crucial aspects such as flexibility, mobility, and technique. These additional factors could potentially affect the relationship between muscle imbalance and injury and therefore require further study.

One more limitation of the study is the lack of a control group. This makes it difficult to determine the causal relationship between muscle imbalance and injury outcomes. Although the study offers valuable insights into the association, it is important to interpret the conclusions carefully since it is not possible to draw definite cause-and-effect conclusions without a control group for comparison.

To overcome these limitations, future research should focus on larger and more diverse samples, consider a wider range of factors, and include appropriate control groups. By doing so, we can gain more comprehensive and robust insights into the connection between muscle imbalance, injury, and the causal factors involved.

### 4.2. Strengths and Practical Implications of the Study

This research provides valuable insights into the complex realm of muscle imbalances among young elite badminton players. A notable strength of this study is the use of a reliable and validated method to measure the strength of shoulder muscles during internal and external rotation. The results are intriguing, highlighting significant differences between the dominant and non-dominant shoulders, as well as variances between male and female players.

Coaches and trainers working with young elite badminton players can benefit greatly from the recent findings on muscle imbalances. By creating customized training programs that address these issues, coaches and trainers can help to reduce the risk of injuries and improve the overall well-being and performance of their players. It is important to note that ongoing attention and monitoring of muscle imbalances is necessary, especially in regard to the dominant and non-dominant shoulders and differences between male and female players. Therefore, targeted training programs that cater to the individual needs of each player should be developed and implemented.

This study has shed light on the dominant side, but it has also raised interesting questions about the non-dominant side and how it relates to injuries in overhead sports. Future research should explore this unchartered territory and uncover the complexities surrounding the non-dominant side and its potential role in preventing injuries among athletes who participate in overhead sports. This study provides coaches and trainers with important information, encouraging them to address muscle imbalances in young elite badminton players. By doing so, they can proactively protect their athletes’ well-being and optimize their performance. In addition, future research should focus on unraveling the mysteries of the non-dominant side, which will contribute to our understanding of injuries in overhead sports.

## 5. Conclusions

In the realm of overhead sports, a key finding of this present study is the identification of muscular imbalances between the shoulders and the trunk in young elite badminton players. Specifically, a notable discrepancy in muscle imbalance was observed between the dominant and non-dominant shoulders in internal and external rotation. This study revealed a significant 27.93% muscle imbalance favoring the dominant shoulder, highlighting the pronounced strength asymmetry in these athletes.

Additionally, when comparing male and female elite badminton players, the study uncovered disparities in strength between their dominant and non-dominant shoulders during rotation. The findings indicate a significant 24.540% muscle imbalance favoring the dominant shoulder in males, while females exhibited a substantial 31.333% muscle imbalance favoring the dominant shoulder. Notably, this reveals a 6.793% higher muscle imbalance in females compared with their male counterparts.

These significant differences emphasize the importance of implementing specialized programs that directly address muscle imbalance issues in elite badminton players. Targeting these imbalances through tailored strength training interventions can mitigate the risk of injuries. Such proactive measures are crucial for sustaining performance and ensuring the longevity of players in the competitive world of badminton.

It is worth noting that the presence of shoulder muscle imbalance significantly increases the likelihood of shoulder injuries. Therefore, the study strongly recommends the introduction of timely intervention programs that focus on rectifying these imbalances. By proactively addressing and alleviating muscle imbalances, athletes can reduce the risk of injuries, thereby safeguarding their performance and long-term success in badminton.

## Figures and Tables

**Figure 1 sports-11-00164-f001:**
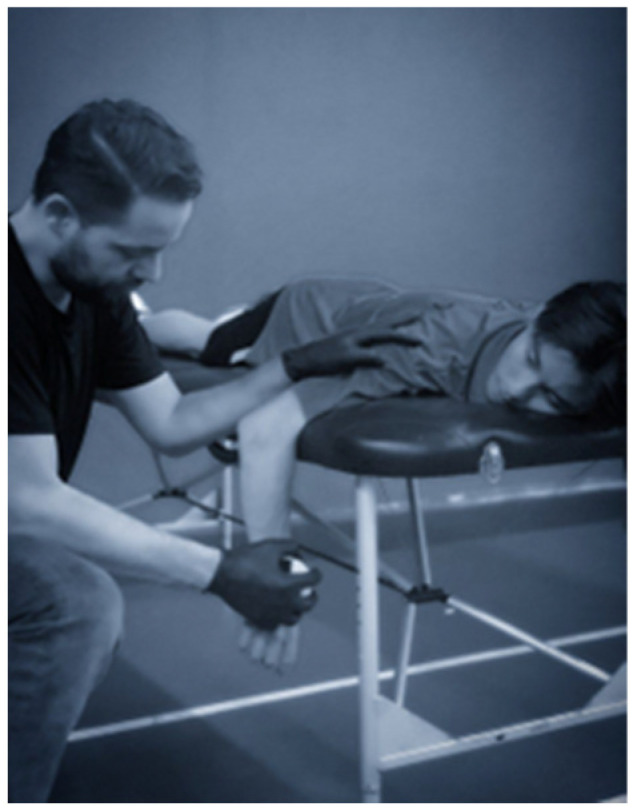
External Rotation Test.

**Figure 2 sports-11-00164-f002:**
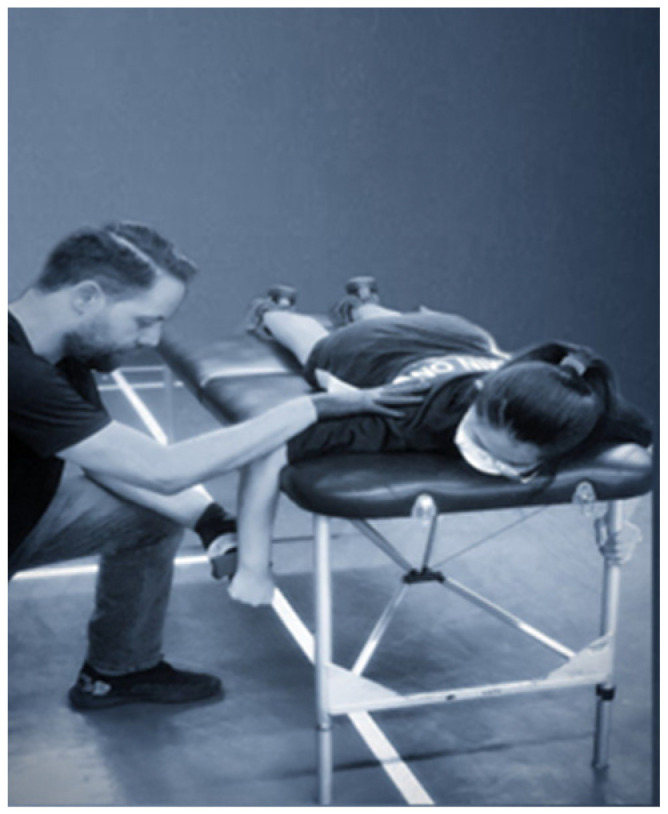
Internal Rotation Test.

**Figure 3 sports-11-00164-f003:**
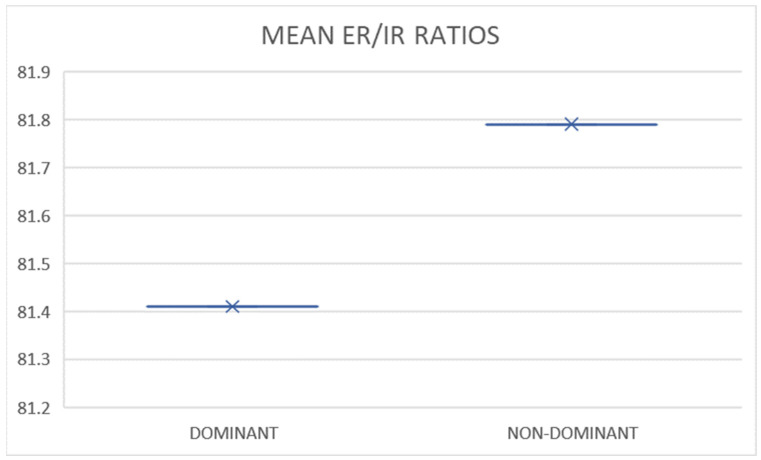
The Differences between Means in Dominant and Non-dominant Ratios of Shoulder Movement in Elite Badminton Players (*p* = 0.87).

**Figure 4 sports-11-00164-f004:**
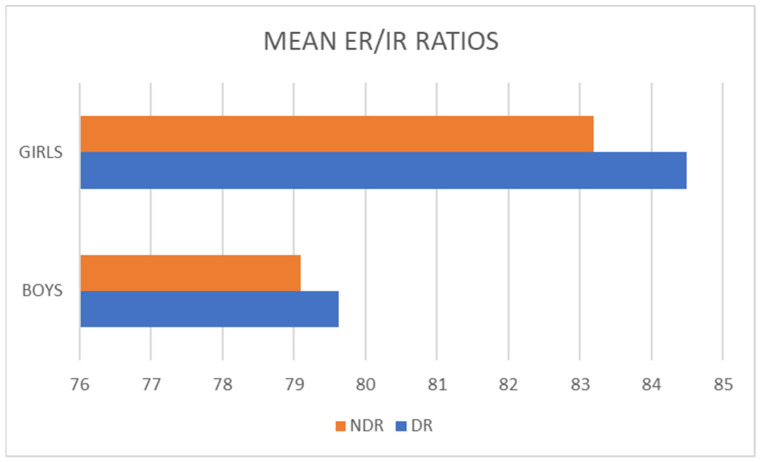
Gender Differences in Dominant and Non-dominant Ratios of Shoulder Movement among Elite Badminton Players in Muscle Imbalance (*p* = 0.000).

**Table 1 sports-11-00164-t001:** Characteristics of Participants (N = 30) (15 males, 15 females).

Characteristics	Mean (Standard Deviation)
Age (year)	17.4 (1.8)
Weight (kg)	56.08 (5.7)
Height (cm)	166.16 (6.6)
Years of playing badminton	7.23 (1.77)
Training per week (h)	16.33 (3.07)

**Table 2 sports-11-00164-t002:** The Differences in Means between Dominant and Non-dominant Shoulder Internal and External Rotators Strength in Elite Badminton Players (Newton).

	Descriptive Statistics Mean Std. Deviation		*t*-Test			
			Difference	*t* Value	*p*-Value	Mean %
DIR	23.75	4.26	6.04	25.4	13.36	0.00
NDIR	17.71	3.78				
DER	19.17	3.31	4.87	25.4	15.18	0.00
NDER	14.30	3.16				
DR (%)	81.41	11.67	0.38	2.7	0.17	0.87
NDR (%)	81.79	14.52				

DIR—Dominant Internal Rotation; NDIR—Non-Dominant Internal Rotation; DER—Dominant Eternal Rotation; NDER—Non-Dominant External Rotation; DR—Dominant Ratio; NDR—Non-Dominant Ratio.

**Table 3 sports-11-00164-t003:** The Differences between Dominant and Non-dominant Ratios of Shoulder Movement.

		F	*p*-Value	*t*	df	Sig. (2-Tailed)	Mean Difference
DR and NDR	Equal variances assumed	132.284	0.000	24.711	166	0.000	27.9368
	Equal variances not assumed			24.711	84.130	0.000	27.9368

DR—Dominant Ratio; NDR—Non-Dominant Ratio.

**Table 4 sports-11-00164-t004:** Gender Differences of Shoulder Internal and External Rotators Mean Strength in Elite Badminton Players (Newton).

	Gender	Descriptive Statistics		*t*-Test			
		Mean	Std. Deviation	Difference		*t* Value	*p*-Value
				Mean	%		
DIR	Boys	25.47	4.71	−3.43	−13.5	−2.38	0.02
	Girls	22.03	3.02				
DER	Boys	20.08	3.17	−1.82	−9.1	−1.54	0.13
	Girls	18.26	3.30				
NDIR	Boys	19.02	4.43	−2.63	−13.8	−1.99	0.06
	Girls	16.39	2.52				
NDER	Boys	14.83	3.53	−1.05	−7.1	−0.90	0.37
	Girls	13.78	2.77				
DR (%)	Boys	79.63	9.83	3.57	4.5	0.83	0.41
	Girls	84.50	13.37				
NDR (%)	Boys	79.09	15.44	5.41	6.8	1.02	0.32
	Girls	83.19	13.52				

DIR—Dominant Internal Rotation; NDIR—Non-Dominant Internal Rotation; DER—Dominant Eternal Rotation; NDER—Non-Dominant External Rotation; DR—Dominant Ratio; —Non-Dominant Ratio.

**Table 5 sports-11-00164-t005:** Gender Differences on Dominant Ratio and Non-dominant Ratio of Shoulder Movement among Elite Badminton Players in Muscle Imbalance.

Gender			F	*p*-Value	*t*	df	Sig. (2-Tailed)	Mean Difference
Male	DR and NDR	Equal variances assumed	118.542	0.000	16.956	82	0.000	24.5404
		Equal variances not assumed			16.956	41.754	0.000	24.5404
Female	DR and NDR	Equal variances assumed	69.362	0.000	20.086	82	0.000	31.3331
		Equal variances not assumed			20.086	41.446	0.000	31.3331

DR—Dominant Ratio; NDR—Non-Dominant Ratio.

## Data Availability

For those interested, the data from this study can be obtained by contacting the corresponding author. However, please note that the data are not publicly accessible due to privacy limitations.

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
