# Peer review of "Gender-Specific Patterns of Muscle Imbalance in Elite Badminton Players: A Comprehensive Exploration"

_sports, 2023, doi:10.3390/sports11090164_

Round 1
Reviewer 1 Report
Dear authors,
please find comments and suggestions below:
Abstract: includes only differences in external rotator strength/internal rotator strength ratios of the dominant and non-dominant sides, not other differences! Should be added!
The style of writing does not seem to be appropriate for a scientific journal as is too "sensationalistic" or too "poetic" since terms as "fascinating" results, " fascinating" study, "clever" approach, "venturing into real of elite badminton players" , "to unreveal the secrets of shoulder movement" etc. are used, to mention only some and all should be corrected.
Title
The present study itself does not analyse injuries related to shoulder muscle imbalance therefore "injuries" should not be included in the title; however gender differences were studied, but not mentioned in the title.
Line 103: instead of "weakness in strength" change to muscle weakness
the goal of the study 103-105
(i) previous studies : "muscle imbalances " in badminton players where studied before and the authors should state precisely which imbalances were studied before and in which population according to the age
(II) present study: What is actually is new in the present study should be added (adolescents, gender differences); please reformulate
The description of the participants: what does "elite" mean since they were "abstaining" from national competitions? Were these recreational players? Suppose not, not clear.
Structure of the article
How external/internal strength ratios were calculated is not "statistical analysis"; it should be moved to "Procedures"
Please add also a brief description of F-test in statistical analysis!
Lines 201-206: The repeated description of the t-test significance under "Results" is unnecessary!
Please use either P- value or p- value.
Lines 226-231 are comments but are presented in section "Results", please move appropriately
Interobserver variability: is described, but without actually using the term (line 157); please correct and present the results of interobserver variability!
Figures, Tables, Graphs
FIG 1 and FIG 2: the back ground is distracting (unnecessary details); in Fig 1. external rotation is hardly visible, must be improved
Table 1:
Title of Table 1: must be changed : Characteristics of sample (N=30) Mean (standard deviation)
Characteristics of participants (N=30); without mean (Standard deviation) in the title:
Please add gender details: number of female and male players
At "Training per week "Units are missing
Table 2:
(i) According to the title "The differences between dominant versus non-dominant …" and since the dominant side is stronger a positive difference exists between dominant and non-dominant, however negative values are in the lines below; the authors should change or the title or the negative values!
(II) No units are stated for "Mean"
(III) First four values (in first four lines) are absolute values; the last two values are % which should be indicated
(IV) Usually the explanation of the abbreviations is direct as DIR= dominant internal rotation ;NDIR= non-dominant internal rotation; etc. DR= (Dominant external rotation/ dominant internal rotation)x 100; NDR=( Non-dominant external rotation/ non-dominant internal rotation) x100;
Please correct in order to make more reader friendly format!
Table 3 and Graph 1:
Table 3 ( units - % are missing for Mean Differences)
Please explain the differences between
DR= (Dominant external rotation/ dominant internal rotation)x 100 presented in table 2 i.e. 81.41% and the value presented in Graph 1 which is around 97% and
NDR=( Non-dominant external rotation/ non-dominant internal rotation) x100 presented in table 2 i.e. 81.79% and the value presented in Graph 1 which is around 70%?
(Dominant external rotation/ dominant internal rotation)x 100 should be always presented with the same abbreviation not as it is now: in Table 2 DR in Table 3 ER/IR Ratio D! The same for non-dominant!
Units should be added to y axis in Graph 1!
The titles of Table 3 and Graph 1 should be improved!
I would recommend not to use muscle imbalance for " ER/IR ratio" since this is only one type of muscle imbalance! (Other imbalances are: between internal rotators of the dominant and non-dominant side and external rotators of dominant and non-dominant side).
Table 4:
The title should be changed to Gender differences of….
New parameter is introduced in Table 4 IRAS (IRAS=internal rotators asymmetry) and ERAS (ERAS=external rotator asymmetry); The authors should explain what is the difference between DIR- minus NDIR and IRAS and DER minus NDER and ERAS!
Units, similar to Table 2 are missing and should be added!
Table 5 and Graph 2:
Table 5: units [%] are missing for Mean Difference
Similar to Table 3 and Graph 1, Table 5 and Graph 2 do not correspond to values in Table 4! Please explain!
Since results for different gender are presented in Graph 2, 4 values (means) should be presented , not two! (DR - boys, NDR- boys, DR - girls, NDR - girls) Please explain and correct.
Units are missing in Y axis in Graph 2.
Titles of table 5 and graph 2 should be modified to Gender differences…
Discussion
line 279: limbs change to dominant and non-dominant shoulder
line 279: 25.4% variance should be changed to 25.4% difference (Table 2).
Mean differences from Table 3 and Table 5 are not discussed and this discussion should be added.
Conclusions
The authors should stress what is new from the present study not repeat the findings of previous studies.
The style of writing does not seem to be appropriate for a scientific journal as is too "sensationalistic" or too "poetic" since terms as "fascinating" results, " fascinating" study, "clever" approach, "venturing into real of elite badminton players" , "to unreveal the secrets of shoulder movement" etc. are used, to mention only some and all should be corrected.
Moderate editing of English language required
Author Response
Dear Reviewer,
I replied, in the attachment, to your requests.
We thank you for the time offered to study our article, but especially for your advice. Your recommendations clarified us even more and helped us to create a much better version of our article.
Also, we checked and corrected the expression in English, with the help of a specialized translator.

Reviewer 2 Report
The following are the issues that are not addressed in this paper.
Abstract
· “gameplay” – specify what kind
· “fascinating results” – give the more modest term
· “muscle imbalance” – define the meaning
· “stronger” – how this is supported by the results
· Provide the hypothesis for this study
Introduction
· “conditions” – change to discomfort or disability
· “postural issues” – what kind?
· “Agonist and antagonist muscles are related, and Codine et al. [14] explored this relationship and showed how it provides insight 73 into the pathological state of the joint.” – rephrase and give details. This is a very general statement
· “disparities” - change to differences
· Lines 75-82 – clarify is these findings related to sports activities and of what kind.
· “An increase in the agonist-antagonist muscle imbalance results from shoulder internal 83 rotation proportionally becoming stronger than its antagonist during long-term badminton training [24].” -= rephrase (very difficult to follow)
· “Athletes should be aware that the internal rotation/external rotation 85 strength of their dominant and non-dominant shoulders may vary significantly when it 86 comes to shoulder stability and injury prevention [25]” – rephrase
· Provide data what are the values of the desired “strength of the shoulder internal rotation/external rotation” for the prevention of injuries
· Provide a clear hypothesis for this study
Methods
· "In this fascinating study” – avoid superlatives
· “cutting-edge” – provide an explanation why this is a cutting-edge device or alternatively avoid this term
· “Microfet 2TM dynamometer” – provide details on the manufacturer and information on the range of the possible measurements, calibration, and errors of measurement readings (technical data on the device).
· “clever” – remove
· “This study promises to shed light on the intriguing dynamics 116 of shoulder strength in badminton players.” – why?
· “Embarking on a journey to uncover the physical prowess of young badminton elites, 119 this captivating cross-sectional study assembled a remarkable group” – rephrase avoiding superlatives
· “30 players” – why this number? Provide statistical justification for the number of included individuals. Provide the inclusion and exclusion criteria
· “around 17 years” – provide average age with a range of ages
· “possessed an impressive combination of agility and strength” - how you defined the agility and strength? If this is a general impression then remove this statement
· “As the anticipation built, it became evident that the majority of the players were right-handed, with only a select few mastering the left-handed technique.” – remove and provide numbers of right-dominant and left dominant individuals.
· “intriguing 7 players” – explain why they are intriguing concerning this study
· “These talented individuals, honing their badminton expertise”, “ensuring they were well-versed in the art of the game.” – rephrase. Please remember that the text should be according to the style of scientific, not fiction, writing.
· “To ensure the study's integrity, strict criteria were set for eligibility, ensuring that the participants were free from upper-limb issues, maintaining a training frequency of no more 131 than four times a week, and abstaining from national competitions.” – rephrase as a part of inclusion and exclusion criteria with reasoning.
· “In the pursuit of accurate data,” – rephrase by mentioning to resolve the factor of muscle fatigue.
· “The players were not only physically prepared but also well-informed about the study's objectives and experimental procedures” – remove, this should be part of informed consent
· “The stage was set for a captivating exploration of the physical capabilities of these extraordinary badminton prodigies.” – remove. This statement doesn’t provide any substantial information for this study.
· Table 1 should be mentioned in the text
· “To unlock the secrets of shoulder strength, a cutting-edge device called the Micro- 153 fet 2TM, Hand-Held Dynamometry (HHD) (Hoggan Health, Draper, UT, USA) was 154 brought into play. In the skilled hands of an assessor well-versed in its techniques, this 155 powerful tool” – the technical data of the dynamometer should be provided in the previous section and the text should be removed for the reasons mentioned above.
· “careful supervision” – by whom?
· “five seconds” – why? Check the publications of Rosenberg N
· “Throughout the arduous process of creating force, the tester provided verbal encouragement, inspiring the participants to push their limits.” – this is non-measurable variable
· “uncovering remarkable capabilities along the way.” – remove or rephrase
· How the differences in anatomy, i.e. the length of the arm and forearm affected the final force readings. Did you standardize the placement of the dynamometer?
· “It is often used in scientific research to determine whether there is a significant difference between the means of two groups . The independent samples t- test assumes that the two samples are independent of each other and that they are 189 normally distributed. The null hypothesis of the independent samples t-test is that there 190 is no significant difference between the means of the two groups, while the alternative 191 hypothesis is that there is a significant difference. e [33]. In summary, the independent samples t-test is a statistical test used to compare the means of two independent groups or 193 samples [34]. It involves stating null and alternative hypotheses, collecting data, calculating the pooled standard deviation and the t-statistic, and comparing it to a critical value to determine if the null hypothesis should be rejected or not [35].” – remove this is obvious. You have to mention why you used these tests in relation to the groups' characteristics
· In order to compare muscle strength data between two groups you should provide some kind of standardization – for example, to lean body mass, otherwise the comparison will include additional parameters ( weight, height, etc) that affect the overall measured strengths and such comparison will be meaningless
· “Venturing into the realm of elite badminton players, the researcher embarked on a quest to unravel the secrets of shoulder movement.” – remove
Results
· “The initial hypothesis (H01) sought to 201 shed light on the disparities between the dominant and non-dominant internal and external rotation of the shoulder among these exceptional athletes.” – define exactly what was the hypothesis of this study and put it in the Introduction
· “To substantiate these differences and understand their significance, an independent t-test was deployed, meticulously comparing the dominant and non-dominant shoulder movements of these badminon elites.” – this might be an objective of the study and if so should be in the Introduction
· “A striking revelation emerged: an astonishing” – remove
· All the Results and Discussion sections are very difficult to follow. Should be completely rewritten and the figures and tables should be clearly related to the text ,the discussion should be related to the hypothesis.
To summarize:
The text should be rewritten according to the standard scientific English style.
Insufficient for a scientific text
Author Response

(The authors gave the same response as above.)

Round 2
Reviewer 1 Report
To the authors
31- instead internal rotation/external rotation change to external rotation/ internal rotation
43- "This may lead" i.e. discomfort does not lead, but is the consequence! To be reformulated
72- "One study's findings" -[23? or cite appropriate reference]
81- "Athletes should acknowledge" change to "Athletes should be acknowledged"
114- "muscle imbalances" add differences in "muscle…"
132- omit years (17.4 (1.8)),
148- Title of Table 1! characteristics of participants, not sample
add units to age
157- unlock the secrets of shoulder strength reformulate
160-162 In the skilled hands of an assessor well-versed in its techniques, this powerful tool measured the internal and external rotation strength of the participant's shoulders. Assisted by a secondary assessor with over three years of experience using the device.
Unite both sentences, reformulate
179 - Figure 2: remove the background (as done in Fig.1)
189- muscle balance change to muscle imbalance
190- calculating external/internal ratios (ER/IRX100%) were calculated add for the dominant side i.e. "dominant ratio (DR) and for the non-dominant side i.e. "non-dominant ratio" (NDR).
210- our subjects. change to participants
Table 2. Title add The differences in means..
add units for strength [ ]
add [%] for DR and NDR
Table 3. Title: add The differences in variances….
Change P-value to p-value
correct misspelling "eternal"
ER/IR Ratio D is the same as DR (external/internal ratio) ; do not use different abbreviations for the same parameter!
ER/IR Ratio N is the same as NDR! Please correct!
Should explain the differences between Table 2 and Table3!
230- …caption of Graph-1: "between…" add between means and at the end add p- value p=0.87
Table4: Title: add mean strength
add units for strength [ ]
and [%] for DR and NDR
Abbreviations: use direct explanation as in table 2!
Table 5: Title add …in variances of Dominant….
Graph 2: is not understandable; from which values, presented where, it was calculated?
Abbreviations: use direct explanation as in table 2!
ER/IR Ratio D is the same as DR (external/internal ratio)
ER/IR Ratio ND is the same as NDR!
Should explain the differences between Table 4 and Table5!
282 - between motor expression replace in strength
283- off in the badminton players included in the study.
287- These determinations of ours replace the results
301- add in older badminton
302- with older age
395- and trunk …elite add young…
397- during replace in
Answer to the questions of the first review:
1. at the new version (lines 162) it is mentioned that the measurements were performed by two assessors (ratters).
Obviously inter-observer (inter-assessor) variability exists.
in line 128 intraclass correlation coefficient (ICC) score of 0.7 with two ratters is mentioned regarding the size of the sample. I am interested in the degree of the variability between the assessors (intraclass correlation coefficient for the two assessors) .
2. I agree that ER/IR ration should be kept!
However, I just wanted to point to the fact, that this ratio is only one type of "imbalances" discovered in this study!
Minor editing required; style of writing improved.
Author Response
Dear reviewer
Thank you for your time and attention to our article. Please see the attachment.

Reviewer 2 Report
The following are still not addressed:
· In the abstract, give the hypotheses and objectives separately
· “we implemented an independent t-test, comparing” – should be changed to “compared”
· “The participants underwent an extensive 105 battery of concentric and eccentric internal and external rotation exercises, which were 106 administered using an isokinetic dynamometer and executed at a consistent speed of 107 60°/s.” – should be in the Methods not in the introduction
· “To unlock the secrets” – change to “to evaluate”
· “to shed light on the intriguing” – change to “ assess “
· “The independent samples t-test is a statistical test used to compare the means of two 193 independent groups or samples. It is often used in scientific research to determine whether 194 there is a significant difference between the means of two groups. The independent samples t-test assumes that the two samples are independent of each other and that they are 196 normally distributed. The null hypothesis of the independent samples t-test is that there 197 is no significant difference between the means of the two groups, while the alternative 198 hypothesis is that there is a significant difference [33]. In summary, the independent samples t-test is a statistical test used to compare the means of two independent groups or 200 samples [34]. It involves stating null and alternative hypotheses, collecting data, calculating the pooled standard deviation and the t-statistic, and comparing it to a critical value 202 to determine if the null hypothesis should be rejected or not [35]. “ – remove , this a common knowledge regarding the statistical test. Just provide information on what statistical test was used and justify this according to the type of variables compared.
· “To address the first hypothesis (H1) we engaged in the evaluation and analysis of 207 specific movements and thus, according to the data presented in the tables we highlighted 208 the existence of differences between the dominant and non-dominant internal and exter- 209 nal rotation of the shoulder among our subjects.” – this should be in the Methods
· “The data presented in Tables 2 and 3, and Graph no. 1 argue, numerically and visually, the significant differences observed in muscle imbalance” – change this general statement to data presentation and refer in brackets to the tables and figures.
· “we initiated 232 a secondary approach to our research approach, and thus, through hypothesis 2, we aimed 233 to determine the existence of muscular imbalances among young male and female elite 234 badminton players.” – rephrase
· “For this we applied the t-test (parametric) for independent samples, 235 to evaluate if there were significant differences in muscle imbalances between the dominant and non-dominant shoulders of the two sexes.” – should be in the Methods
· “24.540%” – 24.54%, “31.333%” – 31.33%
· “By analyzing the difference between the average values, it can be determined that 245 the male badminton players, included in the study, although they have muscle imbalances 246 at the shoulder level, which are 6.793% lower compared to those determined for the fe- 247 male badminton players, confirming hypothesis number 2.” – rephrase
· “The data that support these 248 findings are highlighted in Tables 4 and 5, but also in Graph no. 2.” – how it supports? Provide the data and refer to the tables and figures
· “These determinations of ours indicate that there may be compli- 287 cated and different dynamics of shoulder force manifestation in young badminton player 288 subjects, suggesting unique patterns and potential influences that shape their physical capabilities” – what are the differences you found?
· “The results revealed significant differences between the dominant and non-dominant 298 shoulder, with a significant difference of 25.4% in both internal and external rotator 299 strength.” – differences in what direction?
· “this study corroborated previous findings that players involved in racquet sports possess greater strength in both the internal rotator and external rotator muscles,” – in comparison to whom? It is also contradictory to your previous statement that such data is new.
· “Smash technique” – define
· “When conducting a study, it's crucial to recognize the limitations that come with the 347 results. This specific study has a few limitations that should be noted. To begin with,.” – remove
· “Another important point is that the study only examined the isometric strength of the internal and external shoulder muscles.” – refer to the similar study: Chezar A, Berkovitch Y, Haddad M, Keren Y, Soudry M, Rosenberg N. Normal isometric strength of rotatorcuff muscles in adults. Bone Joint Res. 2013 Oct 7;2(10):214-9. doi: 10.1302/2046-3758.210.2000202. PMID: 24100165; PMCID: PMC3792443.
Additional editing is required
Author Response

(The authors gave the same response as above.)

Round 3
Reviewer 1 Report
Dear authors,
Table 3 and Table 5: I realized that each is actually composed of two tables (one table for F test and one table for t test). I am suggesting that you rearrange the tables 3 and 5 that this would be more visible; in this case you should ignore my remark in review 02 that " in variances" should be added and change the title
Why don't you use the same abbreviations for ratios in table 2 and 3 and table 4 and 5 (DR and NDR)?
Graph 2: add to caption appropriate p values
line 163 newtons - capital letter
Minor editing of English language required
Author Response
Report 1 round 3
Table 3 and Table 5: I realized that each is actually composed of two tables (one table for F test and one table for t test). I am suggesting that you rearrange the tables 3 and 5 that this would be more visible; in this case you should ignore my remark in review 02 that "in variances" should be added and change the title.
Response:
In Table 5 gender differences but in table 3 among all players so we believe that is visible. We ignored the remark in review 02 and we changed.
Thank you for the analysis effort for our article and the time spent in this direction, we sincerely appreciate these things. We ask the reviewer to be more coherent because we do not understand exactly what change we need to make to the respective tables. We tried to change what we understood; we hope that the changes are complete. We are at your disposal in case you still need it.
Why don't you use the same abbreviations for ratios in table 2 and 3 and table 4 and 5 (DR and NDR).
Response:
We have corrected that.
Graph 2: add to caption appropriate p values.
Response:
We have added that.
line 163 newtons - capital letter.
Response:
We have changed that to capital letter.

Reviewer 2 Report
The previous concerns were addressed.
Adequate
Author Response
Thank you for the analysis effort for our article and the time spent in this direction, we sincerely appreciate this effort you made!!!